# New Insights in 9q21.13 Microdeletion Syndrome: Genotype–Phenotype Correlation of 28 Patients

**DOI:** 10.3390/genes14051116

**Published:** 2023-05-21

**Authors:** Alessandro De Falco, Achille Iolascon, Flora Ascione, Carmelo Piscopo

**Affiliations:** 1U.O.C. Genetica Medica, A.O.U. Federico II, 80131 Naples, Italy; ale.deltafi@gmail.com (A.D.F.); achille.iolascon@unina.it (A.I.); 2Dipartimento di Medicina Molecolare di Biotecnologie Mediche, Università degli Studi di Napoli Federico II, 80136 Naples, Italy; 3Department of Molecular Medicine and Medical Biotechnology, University Federico II, 80131 Naples, Italy; 4CEINGE Biotecnologie Avanzate, 80145 Naples, Italy; 5Hospital Directorate, A.O.R.N. “Antonio Cardarelli”, 80100 Naples, Italy; flora.ascione@aocardarelli.it; 6Medical and Laboratory Genetics Unit, A.O.R.N. “Antonio Cardarelli”, 80100 Naples, Italy

**Keywords:** array-CGH, 9q21.13 microdeletion syndrome, *RORB*, *TRPM6*, *PCSK5*, *PRUNE2*, intellectual disability, abnormal eye physiology, severe myopia, brain MRI anomalies, absence seizure, dysmorphic features, genotype–phenotype comparison

## Abstract

The implementation of array comparative genomic hybridisation (array-CGH) allows us to describe new microdeletion/microduplication syndromes which were previously not identified. 9q21.13 microdeletion syndrome is a genetic condition due to the loss of a critical genomic region of approximately 750kb and includes several genes, such as *RORB* and *TRPM6*. Here, we report a case of a 7-year-old boy affected by 9q21.13 microdeletion syndrome. He presents with global developmental delay, intellectual disability, autistic behaviour, seizures and facial dysmorphism. Moreover, he has severe myopia, which was previously reported in only another patient with 9q21.13 deletion, and brain anomalies which were never described before in 9q21.13 microdeletion syndrome. We also collect 17 patients from a literature search and 10 cases from DECIPHER database with a total number of 28 patients (including our case). In order to better investigate the four candidate genes *RORB*, *TRPM6*, *PCSK5*, and *PRUNE2* for neurological phenotype, we make, for the first time, a classification in four groups of all the collected 28 patients. This classification is based both on the genomic position of the deletions included in the 9q21.3 locus deleted in our patient and on the different involvement of the four-candidate gene. In this way, we compare the clinical problems, the radiological findings, and the dysmorphic features of each group and of all the 28 patients in our article. Moreover, we perform the genotype–phenotype correlation of the 28 patients to better define the syndromic spectrum of 9q21.13 microdeletion syndrome. Finally, we propose a baseline ophthalmological and neurological monitoring of this syndrome.

## 1. Introduction

The implementation of array comparative genomic hybridisation (array-CGH) allows us to detect about 1 Kb chromosome aberrations, known as copy number variants (CNVs), which are not detectable by other conventional cytogenetic techniques [1]. CNVs represent an important cause of neurological disorders (e.g., epilepsy, intellectual disability, and autistic behaviour) [2]. This high-resolution tool allows us to describe new microdeletion/microduplication syndromes which were previously not identified.

Boudry-Labis et al. described the 750 kb region of minimal deletion in 9q21.13 locus as the region that encompasses four genes (*RORB*, *TRPM6*, *NMRK1*, *OSTF1*) and two open reading frames (*C9orf40*, *C9orf41*) [3] (Figure 1). 

*RORB* gene, along with *PRUNE2*, *PCSK5,* and *TRPM6*, is a candidate for the neurological phenotype in 9q21.13 microdeletion syndrome [4]. 

*RORB* (OMIM *601972) is a nuclear orphan receptor that regulates neuronal patterning during cortical development [5]. *RORB* (RAR-related orphan receptor β) is expressed in the temporal cortex, as demonstrated on cortical samples from patients with temporal lobe epilepsy and in rat brains [6] and it probably has a role in neuronal cell differentiation [7]. *RORB* gene has two differentially expressed isoforms, RORβ1 and RORβ2 [8]: they have the same DNA-binding domain but different short N-terminal domains. In humans, RORβ2 is expressed predominantly in the retina and in the pineal gland [9], while RORβ1 is mostly expressed in cortex, spinal cord, and in the pituitary gland [10]. In the mouse, RORβ2 is expressed in the pineal gland and the retina while RORβ1 is expressed in the cerebral cortex, thalamus, and hypothalamus [5]. Furthermore, *RORB*-null mice show impairment in several neurological reflexes, and they display several behavioural changes, particularly in sensory processing [11]. In 2016, Rudolf et al. observed that nonsense and missense mutations and CNVs of various sizes involving the *RORB* gene gave rise to *RORB* haploinsufficiency, resulting in a common phenotypic spectrum including intellectual disability, generalised epilepsy, and absence seizures [12]. In this way, they proposed *RORB* as a new candidate gene for neurodevelopmental disorders. In 2020, Sadleir et al. expanded the phenotype associated with *RORB* pathogenic variants describing cases with an overlap of occipital epilepsy and photosensitive genetic generalised epilepsy [13]. 

Another gene included in the critical region of 9q21.13 microdeletion syndrome is *TRPM6* (OMIM *607009) which encodes for a protein expressed in the intestinal and renal epithelial cells [14]. *TRPM6* has an ion channel domain and a protein kinase domain. This protein is essential for magnesium homeostasis and plays an important role in epithelial magnesium transport [14,15]. Loss-of-function mutations in the human *TRPM6* gene induce hypomagnesemia with secondary hypocalcaemia (HSH). Shortages of magnesium and calcium can cause neurological problems which begin in infancy, such as global developmental delay, intellectual disability, failure to thrive and heart failure [16].

*PCSK5* (OMIM * 600488) is expressed in the spinal cord and in the pineal gland, and it is involved in the transmission of neural signals. Chitramuthu et al. studied the zebrafish co-orthologue of the *PCSK5* gene (*PCSK5.1*) and they have discovered it plays distinct roles in developing the brain, endodermal derivatives, and sensory organs [17]. 

*PRUNE2* (OMIM *601972) is expressed in the nervous system (brain, cerebellum, and spinal cord) and it is thought to be involved in neuronal apoptosis [18]. In 2009, *PRUNE2* was proposed as a susceptibility gene for hippocampal atrophy and Alzheimer’s disease [19]. Nectoux et al. suggested that *MECP2* (OMIM * 300005) could repress *PRUNE2* so duplication of *MECP2* can have similar features as the loss of *PRUNE2* [20]. 

Here, we describe a case of a 7-year-old boy with an intellectual disability, speech delay, severe myopia, seizures, and facial anomalies. Array-CGH was performed *in trio* and it highlighted a de novo 8363 Mb 9q21.13 microdeletion. 

We present a literature overview of the 17 patients described before with 9q21.13 microdeletion syndrome and we collect all the 42 DECIPHER patients with a deletion in 9q21.3 locus overlapping with the deletion of our patient. We excluded the DECIPHER patients without clinical information available and the duplicate patients who were described in the literature and also present in DECIPHER database. 

We make a new classification of patients both from literature from DECIPHER based on both the genomic position of the deletions included in 9q21.3 locus deleted in our patient and both gene involvement. In this way, we have a total number of 28 patients divided into 5 groups. We compare the clinical problems, the radiological findings, and the dysmorphic features of each group and of all the 28 patients of our article and we make genotype–phenotype correlation in order to better define the syndromic spectrum of 9q21.13 microdeletion syndrome.

## 2. Material and Methods

### 2.1. Cytogenetics

Genomic DNA, obtained from the proband after obtaining signed informed consent, was isolated from 1.2 mL of ethylenediaminetetraacetic acid (EDTA) peripheral blood lymphocytes using the MagCore extractor system H16 with a MagCore Genomic DNA Large Volume Whole Blood Kit (RBC Bioscience Corp., Taiwan, China). DNA quantity and purity were determined with NanoDrop One (ThermoScientific, Waltham, MA, USA). 

Array-CGH analysis was performed using Oligo/SNP Array-CGH sex match and the following chip was used: Agilent SurePrint 2 × 400 Oligo/SNPs (Agilent Technologies, Santa Clara, CA, USA), using a 250 average resolution and 10 Mb LOH average resolution, following the manufacturer’s protocol. Copy number data were analysed with Cytogenomics 5.0.2 (Agilent Technologies, Santa Clara, CA, USA).

Genomic positions refer to the Human Genome February 2009 assembly (GRCh37/hg19). This platform was composed of 180,000 60-mer oligonucleotide probes with an overall median probe spacing of 13 Kb (11 Kb in Refseq genes). Hybridised slides were scanned with a microarray scanner (Agilent Technologies, Santa Clara, CA, USA), and analysed using Feature Extraction 10.1 and Workbench 6.5.0.1 (Agilent Technologies). Copy number variations (CNVs) were examined if at least three contiguous oligonucleotides presented an abnormal log ratio. CNVs reported in the Database of Genomic Variants [21] (http://projects.tcag.ca/variation/ accessed on 30 April 2023) and in in-house databases of benign CNVs were excluded from further analysis. 

### 2.2. Ethical Consent

Ethical review and approval were waived for this study, dealing with a case report conducted according to clinical practice guidelines. Written informed consent has been approved by the parents’ proband to publish this paper.

### 2.3. Databases and Bioinformatic Tools

This study makes use of data generated by the DECIPHER community (http://decipher.sanger.ac.uk/ accessed on 30 April 2023) [22]. A full list of centres that contributed to the generation of the data is available from https://deciphergenomics.org/about/stats and via email from contact@deciphergenomics.org. Funding for the DECIPHER project was provided by Wellcome (grant number WT223718/Z/21/Z).

We used UCSC Genome Browser on Human February 2009 Assembly hg19 (http://genome.ucsc.edu accessed on 30 April 2023) [23] and Database of Genomic Variants (DGV) [21] We consulted PubMed Central (PMC) archive (https://pubmed.ncbi.nlm.nih.gov/ accessed on 30 April 2023) for literature overview. Franklin by Genoox (https://franklin.genoox.com/clinical-db/home accessed on 30 April 2023) has been used to classify the CNV of our patients according to ACMG guidelines [24].

### 2.4. Workflow of Literature and DECIPHER Database Search 

From the literature, DECIPHER database and our case study, we gathered in total 28 patients. In order to make a better genotype–phenotype correlation, we classify, for the first time in the literature, all the patients into five different groups (Figure 2). 

From PubMed, we found 17 patients affected by 9q21.13 microdeletion syndrome [3,10,25,26,27]. 

From DECIPHER database, we collected 42 patients with the deletion in 9q21.3 locus overlapping with our patient’s deletion. We excluded five DECIPHER patients without available clinical information (279825, 501796, 374605, 385645, 291952) and the thirteen Boudry-Labis’ patients [3], who have been already quoted in the literature search. From the remnant 24 DECIPHER patients with available clinical information, we filtered 14 patients who form Group 4, whose clinical and molecular interest is outside the aim of our article. In this way, we describe ten DECIPHER patients. Our patients, seven patients from Boudry-Labis [3], Genesio’s patient [26], Tuğ’s patient [27] and, four DECIPHER patients compose Group 1 which is made up of 14 patients. The other five patients from Boudry-Labis [3] form Group 2A. The other three DECIPHER cases compose Group 2B. The last seven DECIPHER cases joined with Baglietto’s patient [10], Bartnik’s patient [25] and the last patient from Boudry-Labis [3] form Group 3, which is composed of six patients.

Further information regarding the selection criteria of patients from DECIPHER database is fully explained in Section 3.3.

## 3. Results 

### 3.1. Proband Phenotype

We describe a 7-year-old male who is the third child of healthy non-related parents. His older brothers are, respectively, 14 and 10 years old, and they are in a state of apparent good health and do not have delayed psychomotor development, eye problems or dysmorphism. Our patient was born at the 39th week of gestational age from eutocic delivery. During pregnancy, his mother decided to undergo an invasive prenatal diagnosis with amniocentesis due to advanced maternal age, and the foetal karyotype result was normal (46, XY). Moreover, the obstetric ultrasound assessments were all normal. The birthweight was 2850 g (2nd–9th percentile, appropriate for gestational age), the length was 49 cm (9th–25th percentile), and the head circumference was 35 cm (50th–75th percentile). At birth, no malformations were found, and he did not require resuscitation or perinatal support. The psychomotor developmental milestones were reached late: he started walking at 18 months and speaking at 4 years. Because of psychomotor delay associated with poor participation/language, the tendency to isolate and elusive eye contact, our patient began neuropsychiatric investigations and, when he was 18 months old, he started psychomotricity and speech therapy with good clinical improvements. When he was 2 years old, he presented absence seizures controlled by treatment with valproic acid. The electroencephalogram revealed bilateral middle parieto-temporal paroxysmal anomalies and rapid onset activity in the right temporal region. At the age of 2 years and 2 months, he underwent ophthalmological consultations with evidence of severe myopia which was treated with corrective lenses. When he was 4 years old, he did a brain MRI which showed symmetrical hyperintensity of the peri-trigonal white matter (Figure 3A) and prominent mesial subarachnoid peri-temporo-polar space (Figure 3B,C). 

Furthermore, the metabolic screening with the assessment of levels of amino acids and organic acids, both in serum and urine, showed no significant alterations. Moreover, the auditory brainstem response and the cardiac ultrasounds resulted all normal. Currently, he attends elementary school with a support teacher. At our evaluation, he presents upslanted palpebral fissure, hypertelorism, high palate, long philtrum, wide mouth, thin upper lip vermilion (Figure 4A,B), clinodactyly of the 5th finger, short hands (Figure 4C,D), and sandal gap (Figure 4E). The patient underwent genetic investigations: array-CGH was performed in trio.

### 3.2. Array-CGH and CNV Classification

Array-CGH (assembly: grch37/hg19) which showed a de novo deletion on chromosome 9, in the region 9q21.13q21.31 (75,505,408-83,868,435 bp), extending for approximately 8363 Mb, not present in DGV controls and containing several coding genes: *RORB*, *TRPM6*, *GNAQ*, *PSAT1*, *CEP78*, *VPS13A*, *PCSK5*, *TLE4*, *GNA14*, *CARNMT1*, *OSTF1*, *ALDH1A1*, *ANXA1*, *PCA3*, *GCNT1*, *FOXB2*, *NMRK1*, *PRUNE2*, *RFK*, *NMRK1*. According to ACMG guidelines for CNV [24], 9q21.13q21.31 deletion is classified as pathogenic (score 1) because it contains protein-coding or other functionally important elements (1A criteria), it overlaps with established haploinsufficiency/loss-of-functions sensitive genes or genomic regions (2A criteria), haploinsufficient predictors suggest that at least one gene in the interval is haploinsufficient (2H criteria), the number of protein-coding RefSeq genes wholly or partially included in the CNV region is between 0–24 (3A criteria) and finally there is case–control and population evidence (4L criteria) (Figure 5A).

### 3.3. Classification in Five Groups of All the Patients

Consulting PubMed, the first case with 9q21.13 microdeletion syndrome was reported in 2012 and he presented with epilepsy, eyelid myoclonia, generalised tonic-clonic seizures and autism [25]. The other nine patients were described in 2013 [3]: all the patients had mental retardation, speech delay, epilepsy, and characteristic facial features. In addition to these nine cases, four cases (patients: 2064, 2065, 249623, 249451) with deletions localised within the 9q21.3 locus, with similar clinical phenotypes have been reported in the DECIPHER database [3]. Another case with mild intellectual disability and idiopathic partial epilepsy has also been reported in 2014 [10]. One more case with severe intellectual disability, epilepsy, global developmental delay, dysregulation of platelet aggregation, dysmorphisms, genitalia malformations and hypothyroidism was described in 2015 [26]. In 2018, Tuğ E. et al. described a twenty-two-month-old boy with development delay, absent speech, attention deficit, hyperactivity disorder and dysmorphic craniofacial features (relative macrocephaly, facial asymmetry, frontal bossing, sparse medial eyebrows, hypertelorism, broad base to the nose, smooth philtrum, large mouth, operated cleft lip and wide spaced teeth) [27] (Table 1).

Consulting DECIPHER database, we have found 42 cases with deletions localised or included in the locus 9q21.13 of interest. We excluded five DECIPHER patients without available clinical information (279825, 501796, 374605, 385645, 291952) and the thirteen Boudry-Labis’ patients [3], who have been already quoted in the literature. In this way, we have our case, 17 patients from the literature and 10 DECIPHER cases. To better study 9q21.13 microdeletion syndrome and each specific region, we have decided to stratify all the patients into the following five groups (Table 1) (Table 2): 

The first group (Group 1) is composed of fourteen patients from our case, Genesio [26], Tuğ [27], seven cases from Boudry-Labis [3], and four DECIPHER patients (288874, 322547, 353804, 333517). All these cases share the deletion of our patients from the genomic position 75505408 to 80890936 (in GRCh37), which involves at least the genes *RORB*, *TRPM6*, *PCSK5*, and *PRUNE2.* Thanks to this group, we explore the phenotypes related to very large deletions, which include other genes outside the four previous genes thought to be candidates for the phenotype of 9q21.13 microdeletion syndrome.The second group (Group 2) is composed of eight patients, who have the 9q21.13 deletion, which involves in two different ways the genes *RORB*, *TRPM6*, *PCSK5*, and *PRUNE2.* This group can be divided in two other subgroups:The first one (Group 2A) totally involves the genes *RORB* and *TRPM6* and partially the genes *PCSK5* and *PRUNE2*. It is formed by five patients from Boudry-Labis [3]. Through this group, we want to investigate the phenotypes related to haploinsufficiency of the main four genes of 9q21.13 microdeletion syndrome.The second subgroup (Group 2B) involves only deletions that include the genes *PCSK5* and *PRUNE2*. This subgroup is formed by three DECIPHER patients (277905, 482479, 331471). We explore the phenotypes related to the deletions of these two genes in order to investigate their role as candidate genes for the neurological phenotype in 9q21.13 microdeletion syndrome.The third group (Group 3) is composed of six patients from Bartnik [25], Baglietto [10], one case from Boudry-Labis [3] and three DECIPHER patients (326506, 254951, 327259) who share a 9q21.13 deletion involving part and/or completely the genes *RORB* and *TRPM6.* Here, we investigate the phenotypes due to the haploinsufficiency of the main two genes involved in 9q21.13 microdeletion syndrome.The fourth group (Group 4) is composed of 14 patients from DECIPHER database: 290187, 337563, 480959, 289387, 433930, 359571, 273886, 283459, 258926, 266517, 390449, 283458, 290279, 275259. Their deletions do not contain the four genes *RORB*.

*TRPM6*, *PCSK5*, and *PRUNE2* and, obviously, the critical region of 9q21.13 microdeletion syndrome. Hence, in our work, we do not consider the patients of this group in the total patient count because their interest is outside the aim of this article (Figure 6A–C).

## 4. Discussion

In total, we report 28 patients with 9q21.13 microdeletion syndrome: 17 from the literature, 10 from DECIPHER and our patient (Table 1). We classify them into four groups and each group has a different frequency of clinical and radiological findings, and dysmorphic features (Table 3). For each of these characteristics, we have reported the number of patients who present them (fraction numerator) and we have also marked the number of patients whose features have been investigated (fraction denominator). For this reason, we have different denominators inside each group.

In Group 1, the number of male and female patients is equal (5/13), the karyotype is normal in 62% of the member of this group while it is altered in 15% of them. The altered karyotypes include a balanced translocation [3] and chromothripsis of the 9q21.13 locus [26]. The diagnosis is made between 1 year and 10 months and 16-year-old patients. The most common features of this group are intellectual disability (100%), global development delay (69.2%) and autistic behaviour (61.5%). Abnormal eye physiology is present in 5/13 patients (38.5%), and it includes strabismus, hyperopia, astigmatism, and nystagmus. Brain anomalies at MRI brain are present in 3/13 cases (23.1%) and they contemplate corpus callosum hypoplasia, an incision in the upper segment of the corpus callosum, delayed myelinisation, and arachnoid cyst. Regarding facial dysmorphism, long philtrum is the most frequent feature (57.1%), followed by high palate (42.9%), open mouth (42.9%), thin upper lip vermilion (42.9%), low anterior hairline (28.6%), hypertelorism (28.6%), wide mouth (28.6%) and upslanted palpebral fissure (14.3%). Dysmorphic features are generally described as an abnormal facial shape in DECIPHER case 353804. Other features described in this group are hydrocele testis and short stature (DECIPHER case: 333517), dysregulation of platelet aggregation, female genitalia malformations, and hypothyroidism [26]

In Group 2A, the majority of patients are male (80%) and the diagnosis is made between 8–16-year-old patients. The karyotype is normal in 80% of the member of this group while it is unknown in 20% of them. The most common features of this group are intellectual disability (100%), global development delay (100%), seizures (80%), and autistic behaviour (60%). Abnormal eye physiology is present in 1/5 patients (20%), and it includes strabismus. Brain anomalies at MRI brain are present in 2/5 cases (40%) and they contemplate Arnold-Chiari malformation type I and slight hippocampal asymmetry. Regarding facial dysmorphism, long philtrum is the most frequent feature (57.1%), followed by high palate (42.9%), open mouth (42.9%), thin upper lip vermilion (42.9%), low anterior hairline (28.6%), hypertelorism (28.6%), wide mouth (28.6%), and upslanted palpebral fissure (14.3%). Proportionate short stature is present in the thirteenth patient described in Boudry-Labis [3].

In Group 2B, all the patients are male, and the diagnosis is made between 1–3-year-old patients. The karyotype is normal in all the member of this group. The only two features present in this group are intellectual disability (33.3%) and autistic behaviour (60%). Abnormal eye physiology and brain anomalies at MRI brain are not present. Facial dysmorphisms are not reported in the patients of this group too. Short stature is described in DECIPHER case 331471.

In Group 3, there are exactly three male and three female patients. The karyotype is normal in all the members of this group and the diagnosis is made between 2–15-year-old patients. The most common features of this group are intellectual disability (83.3%), autistic behaviour (50%) and seizures (50%). Abnormal eye physiology is present in 2/6 patients (33%), and it includes vertical nystagmus, strabismus, and myopia. Brain anomalies at MRI brain are not reported. Regarding facial dysmorphism, low anterior hairline and hypertelorism are the most frequent features (both 100%), followed by long philtrum (50%), high palate (50%), thin upper lip vermilion (50%), and upslanted palpebral fissure (50%). 

The majority of the 28 patients affected by 9q21.13 microdeletion syndrome are male (57.1%), and the diagnosis is made between 1–16-year-old patients. The karyotype is normal in 21 patients (75%). The most common features of this group are intellectual disability (89.3%), global developmental delay (57.1%), autistic behaviour (57.1%), and seizures (50%). Abnormal eye physiology is present in 9/28 patients (32.1%) while brain MRI alterations are present in 27.3% of patients. Regarding facial dysmorphism, long philtrum is the most frequent feature (64.3%), followed by thin upper lip vermilion (57.1%), high palate (42.9%), hypertelorism (42.9%), upslanted palpebral fissure (35.7%), and wide mouth (35.7%).

## 5. Conclusions

We describe a patient affected by 9q21.13 microdeletion syndrome with severe myopia and new brain MRI findings. Moreover, we collect 17 patients from a literature search and 10 cases from DECIPHER database with a total number of 28 patients (including our case). In order to better investigate the four candidate genes *RORB*, *TRPM6*, *PCSK5*, and *PRUNE2* for neurological phenotype, we make, for the first time, a classification in 4 groups of all the collected 28 patients. This classification is based both on the genomic position of the deletions included in the 9q21.3 locus deleted in our patient (genomic position in GRCh37: 75505408 to 80890936) and on the different involvement of the four candidate genes. In this way, we compare the clinical problems, the radiological findings, and the dysmorphic features of each group and of all the twenty-eight patients in our article. Furthermore, we perform the genotype–phenotype correlation to better define the clinical spectrum of 9q21.13 microdeletion syndrome.

In particular, Group 1 is composed of patients with large deletions which include other genes outside *RORB*, *TRPM6*, *PCSK5*, and *PRUNE2*, which are thought to be candidates for the neurological phenotype of 9q21.13 microdeletion syndrome. Instead, patients from Group 2A and Group 3 present more in detail the characteristics of 9q21.3 locus because their deletions involve, respectively, *RORB* and *TRPM6*, and partially the genes *PCSK5* and *PRUNE2* (Group 2A), and part and/or completely the genes *RORB* and *TRPM6* (Group 3). Finally, Group 2B involves only deletions which include the genes *PCSK5* and *PRUNE2*: although they are outside the critical region of 9q21.13 microdeletion syndrome, the phenotypes associated with these CNVs allow us to define better the role of *PCSK5* and *PRUNE2.*

Observing the frequencies of the characteristics in Group 2A and in Group 3, intellectual disability, autistic behaviour, and seizure are the most specific findings of the involvement of *RORB* and *TRPM6* in 9q21.13 microdeletion syndrome. Conversely, hypotonia is not so frequent (4/28 patients) as it was reported before in the literature (4/13) [27].

Moreover, abnormal eye physiology is frequent in all the groups except for Group 2B, and it is mostly present in Group 1 (38.5%) and Group 3 (33.3%), confirming the importance of the involvement of *RORB* and *TRPM6* for the ocular phenotype. Particularly, our patient presents severe myopia, reported in only one patient (DECIPHER case 254951). These frequencies show us how ocular problems are relevant in 9q21.13 microdeletion syndrome. For this reason, it is important to evaluate baseline ophthalmological monitoring in all the affected patients.

Furthermore, the brain MRI alterations are present in 40% of patients from Group 2A and in 23.1% of patients from Group 1. Chiari type I malformation, hippocampal asymmetry, hypoplasia of corpus callosum, delayed myelinisation, and arachnoid cyst are reported [3]. Our patient’s MRI findings have never been described before in 9q21.13 microdeletion syndrome and they include symmetrical hyperintensity of the peri-trigonal white matter and prominent mesial subarachnoid peri-temporo-polar space. We can therefore highlight how fundamental the periodical neurological monitoring is, and the evaluation of baseline brain MRI in all the patients affected by 9q21.13 microdeletion syndrome.

Regarding dysmorphic features, patients from Group 3 present higher frequencies among the other groups: low anterior hairline and hypertelorism are the most common features (both 100%), followed by long philtrum (50%), high palate (50%), thin upper lip vermilion (50%), and upslanted palpebral fissure (50%). We think *RORB* and *TRPM6* could be responsible for the dysmorphic features of 9q21.13 microdeletion syndrome.

In summary, we describe a rare syndrome in which the main clinical features could be most likely caused by the loss of *RORB* and *TRPM6*, which is deleted in patients affected by 9q21.13 microdeletion syndrome. 

The genotype–phenotype comparison of all the reported 28 patients reveals several common key features (such as intellectual disability, autistic behaviour, seizures, abnormal eye physiology, and brain anomalies), but also a great phenotypic heterogeneity. 

The description of further patients with the deletion in 9q21.13 locus and the clinical updating of the already described 28 patients is desirable to ensure an adequate and targeted follow-up of this very peculiar and rare syndrome, to monitor its continuous clinical evolution and to evaluate a proper follow-up.

## Figures and Tables

**Figure 1 genes-14-01116-f001:**
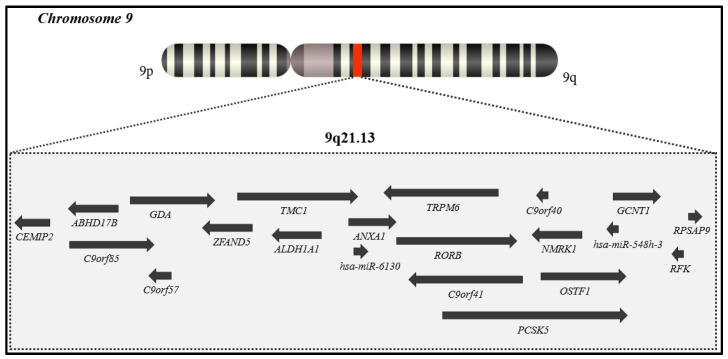
Schematic representation of genes present in the 750 kb region of minimal deletion in 9q21.13 locus (genomic positions in GRCh37: 77047469-77807140). This region encompasses four genes (*RORB*, *TRPM6*, *NMRK1*, *OSTF1*) and two open reading frames (*C9orf40*, *C9orf41*) as described by Boudry-Labis et al [3].

**Figure 2 genes-14-01116-f002:**
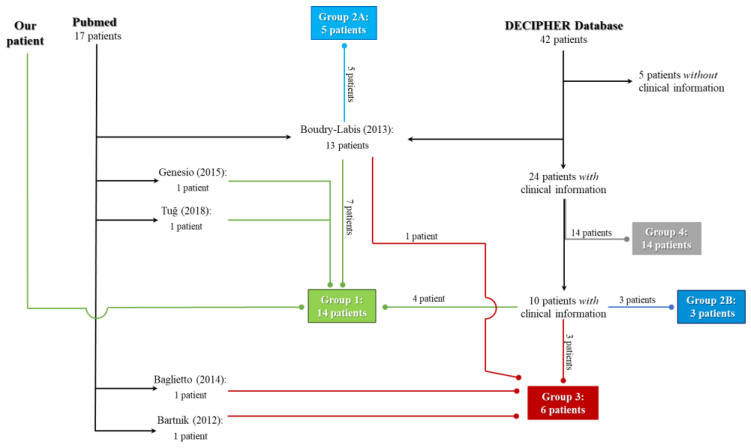
Workflow of literature [3,10,25,26,27] and DECIPHER database search. Normal end arrow indicates where the patients come from (literature or DECIPHER database). The different colour of oval end arrow indicates the belonging group of the patients. Arcs are present where there is a graphical intersection between the arrows.

**Figure 3 genes-14-01116-f003:**
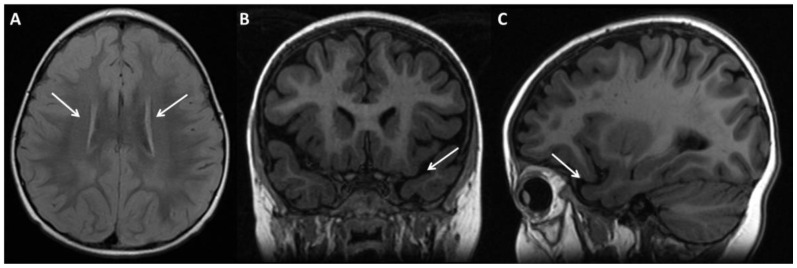
Brain MRI in T2 FLAIR of our patient. (**A**). Symmetrical hyperintensity of the peri-trigonal white matter (as indicated by the white arrows. (**B**–**C**). Prominent mesial subarachnoid peri-temporo-polar space (as indicated by the white arrows). In axial (**A**), coronal (**B**) and sagittal (**C**) images.

**Figure 4 genes-14-01116-f004:**
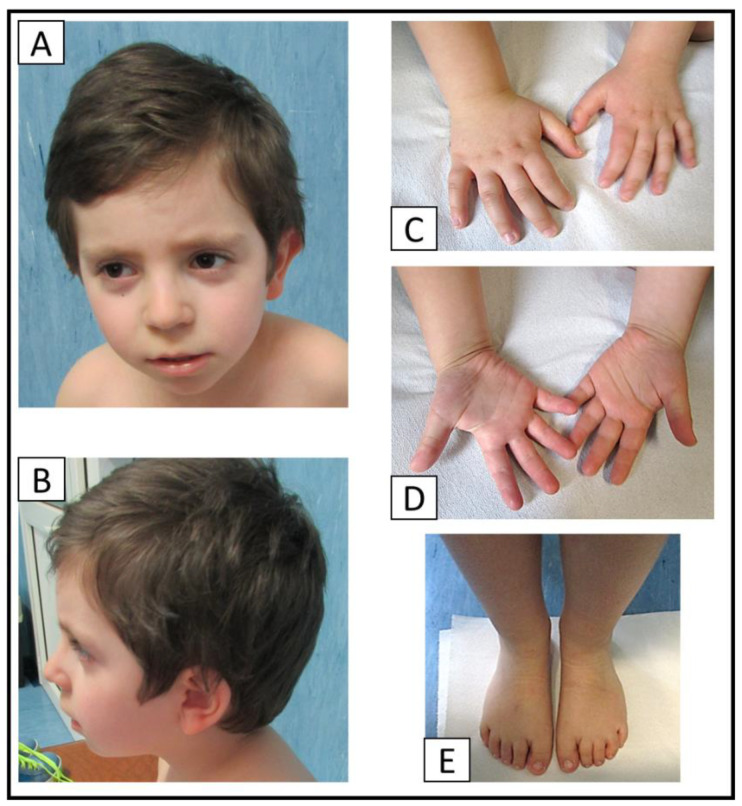
At our physical evaluation, our patient, affected by 9q21.13 microdeletion syndrome, presents upslanted palpebral fissure, hypertelorism, high palate, long philtrum, wide mouth, thin upper lip vermilion (**A**,**B**), clinodactyly of the 5th finger, short hands (**C**,**D**) and sandal gap (**E**) (Figure 4).

**Figure 5 genes-14-01116-f005:**
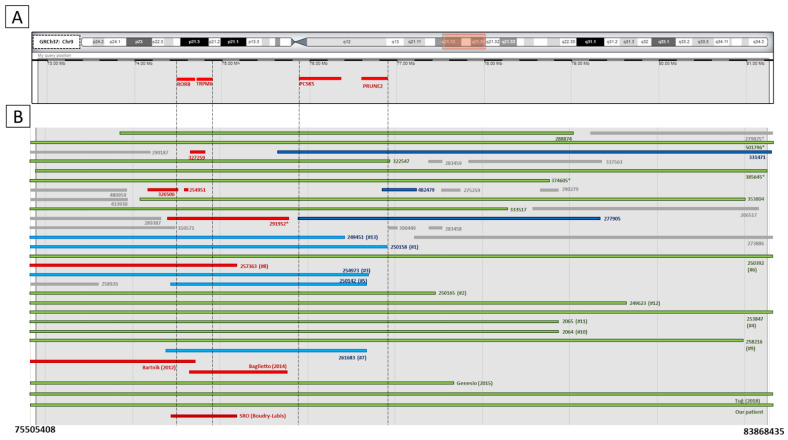
(**A**). Schematic representation of main four genes (*RORB*, *TRPM6*, *PCSK5*, *PRUNE*) present in the de novo deletion on chromosome 9 of our patient; (**B**). A total of 42 cases from DECIPHER genome browser with deletions localised or included in the locus 9q21.13q21.31 of interest. These 42 patients are divided into 5 groups according to the classification proposed in the text and they are represented in different coloured bars: green bars represent deletions belonging to Group 1; light blue bars stand for Group 2A while dark blue bars are referred to Group 2B; red bars represent Group 3 and grey bars are referred to Group 4. Numbers in bold below represent the genomic position of our patient’s deletion: arr[GRCh37]9q21.13q21.31 (75505408_83868435)x1. The vertical dashed lines show genomic regions including the genes of interest. SRO stands for “shortest region of overlap” as defined by Boudry-Labis [3,10,25,26,27]. “#” followed by a number (1–13) stands for the 13 patients from Boudry-Labis [3]. “*” represents the DECIPHER cases without clinical available information. Adapted from DECIPHER genome browser.

**Figure 6 genes-14-01116-f006:**
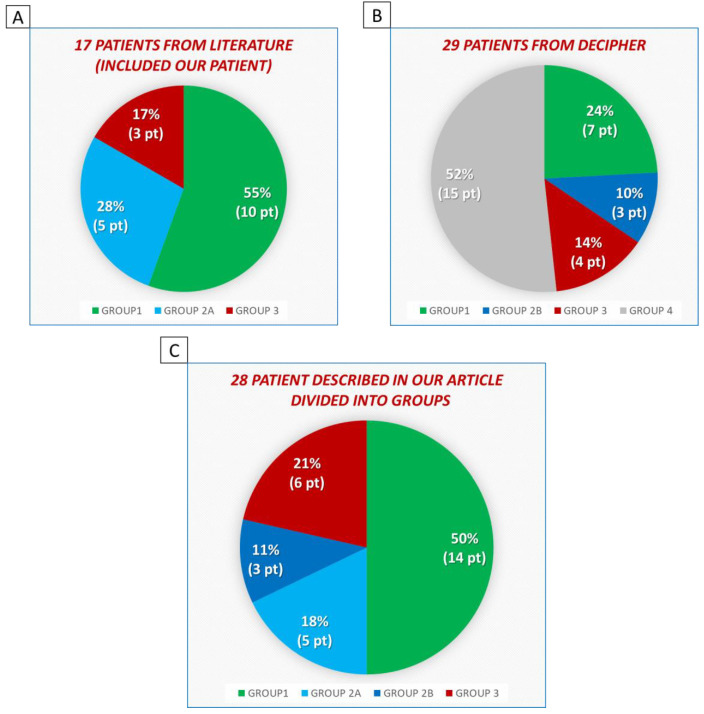
Different highlights of patients affected by 9q21.13 microdeletion syndrome from literature and from DECIPHER database. (**A**). Seventeen literature patients were sorted into their belonging groups. All DECIPHER cases are divided into five groups showing their numerosity in percentages. (**B**). DECIPHER cases, both with and without clinical information, which were never described before, are divided according to their belonging group. The aim of our investigation is to focus on DECIPHER cases with 9q21.13 microdeletion syndrome with clinical information never described before. To do this, we do not consider Group 4 because its deletions do not include the critical region of 9q21.13 microdeletion syndrome. (**C**). Twenty-eight patients were described in our article and divided into the four groups of interest where genotype–phenotype comparison is performed.

**Table 1 genes-14-01116-t001:** The literature [3,10,25,26,27] and DECIPHER cases affected by 9q21.13 microdeletion syndrome were the number of patients from each source, the genomic position in GRCh37, the number of DECIPHER cases (if present) and the belonging group. Note: patients #10, #11, #12, and #13 are referred to by Boudry-Labis as “DECIPHER patients” [3].

Source	Patients (28)	Coordinates of Deletion on Chromosome 9 (Grch37/hg19)	Decipher Patient	Group
Bartnik (2012)	1	74741400-77306932 *	-	3
Baglietto (2014)	1	77254734-78354734	-	3
Boudry-Labis (2013)	13	73920074-79528971 (#1)	250158	2A
		73588788-80076668 (#2)	250165	1
		72182955-79312306 (#3)	254973	2A
		73661807-83532389 (#4)	253847	1
		77047469-79291332 (#5)	250142	2A
		74391472-85348840 (#6)	250392	1
		77058421-79277191 (#7)	261683	2A
		71025196-77807140 (#8)	257363	3
		70950015-83592446 (#9)	258216	1
		75091854-81486732 (#10)	2064	1
		71128855-81486732 (#11)	2065	1
		71128848-82257009 (#12)	249623	1
		71128848-79023977 (#13)	249451	2A
Genesio (2015)	1	72803705–80243747	-	1
Tuğ (2018)	1	71069763-86333272	-	1
DECIPHER	4	76474486-81651005	288874	1
		70984481-79549501	322547	1
		76698559-83614583	353804	1
		73307216-80890936	333517	1
	3	76792356-77124539	326506	3
		77206264-77240837	254951	3
		77271754-77441321	327259	3
	3	78504896-81960668	277905	2B
		79467871-79853516	482479	2B
		78276042-84032536	331471	2B
Our patient	1	75505408-83868435		1

“*” was in NCBI36/hg18 and it has been lifted over in GRCh37 (LiftOver from USCS website, https://genome.ucsc.edu/cgi-bin/hgLiftOver accessed on 30 April 2023).

**Table 2 genes-14-01116-t002:** Table with the literature and DECIPHER cases: in the columns, they are divided into the five groups and the total number of patients; in the rows, they are divided according to their description in literature, the presence of information about them on DECIPHER and the absence of their description in literature, and the patients described in our article. We describe 28 patients excluding group 4 because their interest is outside the aim of our research.

	G.1	G.2A	G.2B	G.3	G.4	TOT
Patients from literature (included our case)	10	5	0	3	0	**17**
Patients from DECIPHER	7	0	3	4	15	**29**
Patients *without* clinical information	3	0	0	1	1	**5**
Patients *with* clinical information	4	0	3	3	14	**24**
**Patients described in our article**	**14**	**5**	**3**	**6**	**-**	**28**

“G.” means “Group”. “TOT” refers to the sum of the patients in the previous rows of the five groups.

**Table 3 genes-14-01116-t003:** Clinical problems, radiological findings, and dysmorphic features of all 28 patients affected by 9q21.13 microdeletion syndrome. In the rows, they are divided into four groups and for each one, there are the frequencies of the indagated characteristics and their percentages. Then, we compare the 27 patients to our patient (grey column), and we recalculate the new percentages of the features of 9q21.13 microdeletion syndrome for the 28 patients (including our case).

Patients with 9q21.13 Microdeletion Syndrome		27 Patients (DECIPHER + Literature) Divided in Four Groups										Our Patient	28 Patients of Our Article	
		G. 1 (13 pt)	%	G. 2A (5 pt)	%	G. 2B (3 pt)	%	G. 3 (6 pt)	%	Total: 27 pt	%	G.1 (14th pt)	Total: 28 pt	%
	**Sex:**											Male		
	Male	5/13	38.5	4/5	80	3/3	100	3/6	50	15/27	55.6		16/28	57.1
	Female	5/13	38.5	0/5	0	0/3	0	3/6	50	8/27	29.6		8/28	28.6
	Unknown	3/13	23	1/5	20	0/3	0	0/6	0	4/27	14.8		4/28	14.3
	**Karyotype:**											Normal		
	Normal	8/13	62	3/5	60	3/3	100	6/6	100	20/27	74.1		21/28	75
	Altered	2/13	15	1/5	20	0/3	0	0/6	0	3/27	11.1		3/28	10.7
	Unknown	3/13	23	1/5	20	0/3	0	0/6	0	4/27	14.8		4/28	14.3
	Age at diagnosis											7y		
	Minimum age	1y 10m		8y		1y		2y		1y			1y	
	Maximum age	16y		16y		3y		15y		16y			16y	
**Clinical and** **radiological findings**	Intellectual disability (HP: 0001249)	13/13	100	5/5	100	1/3	33.3	5/6	83.3	24/27	88.9	+	25/28	89.3
	Global development delay (HP: 0001263)	9/13	69.2	5/5	100	0/3	0	1/6	16.7	15/27	55.6	+	16/25	57.1
	Autistic behaviour (HP: 0000729)	8/13	61.5	3/5	60	1/3	33.3	3/6	50	15/27	55.6	+	16/28	57.1
	Seizure (HP: 0001250)	6/13	46.2	4/5	80	0/3	0	3/6	50	13/27	48.1	+	14/28	50
	Hypotonia (HP: 0001252)	2/13	15.4	1/5	20	0/3	0	1/6	16.7	4/27	14.8	−	4/28	14.3
	Abnormal eye physiology (HP: 0012373)	5/13	38.5	1/5	20	0/3	0	2/6	33.3	8/27	29.6	+	9/28	32.1
	Brain anomalies (MRI brain) (HP: 0410263)	3/13	23.1	2/5	40	0/0	0	0/3	0	5/21	23.8	+	6/22	27.3
**Dysmorphic features**	Low anterior hairline (HP: 0000294)	2/7	28.6	0/4	0	0/0	0	2/2	100	4/13	30.8	−	4/14	28.6
	Hypertelorism (HP: 0000316)	2/7	28.6	1/4	25	0/0	0	2/2	100	5/13	38.5	+	6/14	42.9
	Upslanted palpebral fissure (HP: 0000582)	1/7	14.3	1/4	25	0/0	0	1/2	50	4/13	30.8	+	5/14	35.7
	High palate (HP:0000218)	3/7	42.9	0/4	0	0/0	0	1/2	50	5/13	38.5	+	6/14	42.9
	Long philtrum (HP: 0000343)	4/7	57.1	1/4	25	0/0	0	1/2	50	8/13	61.5	+	9/14	64.3
	Open mouth (HP: 0000194)	3/7	42.9	0/4	0	0/0	0	0/2	0	3/13	23.1	−	3/14	21.4
	Wide mouth (HP: 0000154)	2/7	28.6	1/4	25	0/0	0	0/2	0	4/13	30.8	+	5/14	35.7
	Thin upper lip vermilion (HP: 0000219)	3/7	42.9	3/4	75	0/0	0	1/2	50	7/13	53.8	+	8/14	57.1

“+” stands for clinical features of 9q21.13 microdeletion syndrome present in our patient. “−” stands for clinical features of 9q21.13 microdeletion syndrome not present in our patient. “G.” means “Group”. “%” refers to the percentages calculate on the basis of the frequencies. “y” stands for year(s). “m” stands for months. “pt” means patient(s). “HP” refers to human phenotypes [28].

## Data Availability

Not applicable.

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
