# Peer review of "New Insights in 9q21.13 Microdeletion Syndrome: Genotype–Phenotype Correlation of 28 Patients"

_genes, 2023, doi:10.3390/genes14051116_

Round 1

Reviewer 1 Report

The manuscript of De Falco et al. presents a new case of  9q21.13 microdeletion syndrome and reviews the 24 cases from the literature and DECIPHER database. Although the patient presents with quite a typical phenotype, apart from myopia, given an extensive literature review provided, I find it a valuable manuscript.

Comments:

1.  Abstract is not very informative and quite general. I would add more details e.g. regarding myopia or details on genetic background of all cases

2.  Introduction is very short. On the other hand, in the result section there is detailed description of genes involved. I would transfer the description of genes  e.g. to the introduction or to the discussion

3. Methods: how was the literature search done? what is the website of DECIPHER database? A workflow showing literature search and DECIPHER database search could be useful

4. Discussion: Please comment on the genetic-phynotypic correlation according to the group of the patients.

5. Is brain MRI of the patient available? It would be very valuable to add it.

6. Figure 1 A- I do not find it very informative, it could be omitted.

7. Table 1-Are there any cases where there is no phenotypic information on the feature provided? Now it seems that in all the cases there is a clear yes or no for the reported feature. If there are any cases with no info on the phenotypic feature please highlight.

8. I think that HPO used only in the Table 1 would be better. It could be omitted in the main text.

9. English may need some editing.

Reviewer 2 Report

This study is presented as a case report about patient with rare chromosomal microdeletion at 9q21.13 referred in 2013 as a novel microdeletion syndrome. At the same time, the final part of the manuscript is devoted to analysis of clinical features that together has a significant impact on accumulation of information about clinical manifestation of rare CNV in human populations. Importantly, authors reported about never described before MRI findings in theirs patient. However, some minor revisions are required before accepting.

1.       The abstract must be improved by including major findings from the clinical reports and literature review.

2.       I am not sure, that “9MS” is a good choice for abbreviation. Firstly, other clinically significant microdeletions on chromosome 9 may have the same non-unique abbreviation (for example del9q34.3, known as Kleefstra syndrome-1, MIM#610253). Secondly, “9MS” can be understand also as a microduplication syndrome.

3.       Please, use array-CGH instead of CGH-array in the text.

4.       English should be improved by native speakers. For example, “He performed CGH-array…” Who is performed? Patient? The same is on the page 2: “Our patient and his parents also performed CGH-array…”. etc.

5.       Please, explain for which reasons two types of arrays with 400K and 180K resolution were used in the study.

6.       On the page 2 indicated that “Written informed consent has been approved from the patient(s) to publish this paper”.  I suggest that IC was obtained from parents. The correct information is given in the Informed Consent Statement.

7.       Please add manufacturer (RBCBioscience) for the Automated Nucleic Acid Extractor Mag Core.

8.       Please provide more detailed information about number of previous pregnancies in the woman from reported family and their outcomes.

9.       The demonstration of genetic heterogeneity of 9q21.13 microdeletion syndrome is one of the interesting and important results of the study. Can you designate the specific clinical features to each cytogenetic variant? Please try to perform correspondence analysis for all reported patients to strength the conclusion about genotype-phenotype correlations.

10.   The table A and diagram C on the Figure 2 contain the same information. One of them, table or diagram, can be deleted without loss of the sense.  

Please, see my comments about English in the main part of review.

 I believe, that manuscript should be considered for publication in the "Genes".

Round 2

Reviewer 1 Report

Thank you for the revisions and a detailed answer to my questions.